# Amortized Learning of Flexible Feature Scaling for Image Segmentation

## Abstract

Convolutional neural networks (CNN) have become the predominant model for image segmentation tasks. Most CNN segmentation architectures resize spatial dimensions by a fixed factor of two to aggregate spatial context. Recent work has explored using other resizing factors to improve model accuracy for specific applications. However, finding the appropriate rescaling factor most often involves training a separate network for many different factors and comparing the performance of each model. The computational burden of these models means that in practice it is rarely done, and when done only a few different scaling factors are considered.

In this work, we present a hypernetwork strategy that can be used to easily and rapidly generate the Pareto frontier for the trade-off between accuracy and efficiency as the rescaling factor varies. We show how to train a *single* hypernetwork that generates CNN parameters conditioned on a rescaling factor. This enables a user to quickly choose a rescaling factor that appropriately balances accuracy and computational efficiency for their particular needs. We focus on image segmentation tasks, and demonstrate the value of this approach across various domains. We also find that, for a given rescaling factor, our single hypernetwork outperforms CNNs trained with fixed rescaling factors.

## 1 Introduction

Convolutional Neural Networks (CNNs) have become ubiquitous in computer vision (Khan et al., 2020). Driven by the availability of massive datasets, popular models use large numbers of parameters to exploit the growing amount of information (He et al., 2016; Tan & Le, 2019). These models require large amounts of computation, most often necessitating hardware accelerators like GPUs or TPUs for training and inference (Jouppi et al., 2017). The computational burden at training time increases infrastructure costs and at inference time it limits deployment, specially in resource-constrained applications (Sze et al., 2017; Yang et al., 2017). In turn, the increase in computation induced by large machine learning models leads to increased energy requirements whose associated carbon emissions have a considerable environmental impact (Patterson et al., 2021).

These concerns have led researchers to explore a variety of techniques for minimizing computational requirements while maintaining accuracy. Common strategies include quantization (Hubara et al., 2016; Jacob et al., 2018; Rastegari et al., 2016), pruning (Blalock et al., 2020; Gale et al., 2019; Han et al., 2016; Janowsky, 1989), and factoring (Chollet, 2017; Jaderberg et al., 2014; Xie et al., 2017), which focus on reducing the parameter count and the size of the convolutional kernels. These methods usually introduce a trade-off between model accuracy and efficiency and require training multiple models in order to select the best model based on considerations regarding both metrics. These strategies do not explore reducing the spatial dimensions of the inputs to convolutional layers as a way to reduce the cost of convolutions.

Most modern CNN architectures repeatedly reduce the spatial scale of the features in their architectures by using spatial pooling operations such as max-pooling or strided convolutions. In practice, most network designs downscale by a factor of two (He et al., 2016; Huang et al., 2017; Krizhevsky et al., 2012; Tan & Le, 2019), without considering other possibilities. This can be inefficient, since the amount of rescaling of

**Hyperparameter Search Approach**

**Proposed Amortized Learning with Hypernetworks**

1. Train N networks with varying rescaling amounts independently

2. Choose from trained models or train more using a narrower range

1. Train single hypernetwork conditioned on rescaling

2. Rapidly characterize Pareto frontier of models

CNN with $\varphi = 0.1$
CNN with $\varphi = 0.2$
CNN with $\varphi = 0.3$
$\bullet \ \bullet \ \bullet$
CNN with $\varphi = 0.9$

Accuracy

Inference Cost $= f(\varphi)$

Rescaling Factor $\varphi$

Hypernetwork

CNN with $\varphi$

Accuracy

Inference Cost $= f(\varphi)$

Figure 1: Strategies for exploring the accuracy-efficiency trade-off of varying the CNN rescaling factor. A standard approach (*left*) requires training N models with different rescaling factors independently. Our proposed amortized learning strategy (*right*) exploits the similarity between the tasks and trains a single hypernetwork model. Once trained, we can rapidly evaluate many rescaling settings and efficiently characterize the Pareto frontier between segmentation accuracy and computational cost for each dataset or task of interest. This enables rapid and precise model choices depending on the desired trade-off at inference.

intermediate features plays a major role in network efficiency, because the cost of convolution operations is proportional to the spatial dimensions of the input feature maps.

Some research does explore alternative rescaling strategies (Graham, 2014; Kuen et al., 2018; Lee et al., 2016; Yu et al., 2014; Zhang, 2019) or very recently optimizing the rescaling layers as part of training (Jin et al., 2021; Liu et al., 2020; Riad et al., 2022). However, these techniques either do not explore multiple rescaling factors, or jointly optimize the amount of rescaling while training for a fixed, pre-defined metric. Such approaches do not fully characterize the trade-off between model accuracy and the amount of rescaling performed in rescaling layers.

In this work, we propose an alternative approach. We show how to train a single hypernetwork that captures a complete landscape of models covering a continuous range of rescaling ratios. Once trained, the hypernetwork rapidly predicts model weights for an inference network for a desired rescaling factor. This predicted network is as efficient at inference as a regular network trained for a single rescaling factor. Training the hypernetwork requires scarcely more computation than training a model for a single rescaling factor, leading to a substantial improvement compared to training many standard models. Moreover, somewhat surprisingly, we find that its accuracy is typically better than that of a network trained with a single fixed rescaling factor. Our hypernetwork model can be used to rapidly generate the Pareto frontier for the trade-off between accuracy and efficiency as a function of the rescaling factor (Figure 1). This enables a user to choose, depending upon the situation, a rescaling factor that appropriately balances accuracy and computational efficiency.

We are motivated by the prevalent task of semantic segmentation, and evaluate the proposed method on semantic segmentation tasks on both natural images and medical brain scans. In our experiments, we find that a wide range of rescaling factors achieve similar accuracy results despite having substantially different inference costs. We demonstrate that the proposed hypernetwork based approach is able to learn models for a wide range of feature rescaling factors, and that inference networks derived from the hypernetwork perform at least as well, and in most cases better than, networks trained with fixed rescaling factors.

Our main contributions are:

- We introduce a hypernetwork model that, given a rescaling factor, generates the weights for a segmentation network that uses that amount of rescaling in the rescaling layers.

- We show that our approach makes it possible to characterize the trade-off between model accuracy and efficiency far faster and more completely than than previous approaches, which most often involve training multiple models.

- We use this to approach to demonstrate that our method enables choosing rescaling factors during inference based on rapidly computing an accuracy-efficiency trade-off for a dataset of interest. Since for various segmentation tasks a wide range of rescaling factors lead to similar final model accuracy, this offers the opportunity to substantially improve efficiency without sacrificing accuracy, enabling deployment in more resource-constrained settings, and lowering the energy requirements and carbon emissions of performing model inference.

- We show that learning a single model with varying rescaling factors has a regularization effect that makes networks trained using our method consistently more accurate than networks with a fixed rescaling factor.

## 2 Related Work

### 2.1 Resizing in Convolutional Neural Networks

Most modern CNNs are built from three main building blocks: learnable convolutional filters, normalization layers, and deterministic resizing layers to downscale or upscale features maps (He et al., 2016; Radosavovic et al., 2019; Tan & Le, 2019; Zoph et al., 2018; He et al., 2016). Resizing layers have been implemented in a variety of ways, including max pooling, average pooling, bilinear sampling, and strided convolutions (Bengio et al., 2017). A common drawback of these pooling operations is that they enforce a fixed scale and receptive field on each resizing step. Pooling alternatives, such as strided and atrous convolutions (Chen et al., 2017; 2018; Springenberg et al., 2014) and learnable downsampling in the network (Etmann et al., 2020), present the same limitation of operating at discrete integer steps.

Some alternative pooling approaches improve final model accuracy, such as combining max pooling and average pooling using learnable parameters (Lee et al., 2016; Yu et al., 2014), pre-filtering features with antialiasing filters (Zhang, 2019) or combining features at multiple resolutions (He et al., 2015b). These strategies focus on improving model accuracy and do not thoroughly study the efficiency aspect of the downsampling operations. Other methods have explored a wider range of rescaling operations by stochastically downsampling intermediate features (Graham, 2014; Kuen et al., 2018) leading to improvements to model generalization. Recent work has investigated differentiable resizing modules that replace pooling operations (Jin et al., 2021; Liu et al., 2020), strided convolutions (Riad et al., 2022) or the resizing for preprocessing the input to the model (Talebi & Milanfar, 2021). These techniques optimize the differentiable modules as part of the training process, leading to improvements in model performance for a particular metric. In contrast, our goal is to efficiently learn many models with varying rescaling factors that characterize the entire trade-off curve between accuracy and efficiency. This allows users to choose at test time a model that appropriately balances efficiency and accuracy.

### 2.2 Hypernetworks

Hypernetworks are neural network models that output the weights for another neural network (Ha et al., 2016; Klocek et al., 2019; Schmidhuber, 1993). Hypernetworks have been used in a range of applications such as neural architecture search (Brock et al., 2017; Zhang et al., 2018), Bayesian optimization (Krueger et al., 2017; Pawlowski et al., 2017), weight pruning (Liu et al., 2019), continual learning (von Oswald et al., 2019), multi-task learning (Serrà et al., 2019) meta-learning (Zhao et al., 2020) and knowledge editing (De Cao et al., 2021). Recent work has used hypernetworks to perform hyperparameter optimization (Lorraine & Duvenaud, 2018; MacKay et al., 2019; Hoopes et al., 2022; Wang et al., 2021). These gradient-based optimization approaches train a hypernetwork to predict the weights of a primary network conditioned on a hyperparameter value. Our work is also similar to hypernetwork based neural architecture search techniques in that both explore many potential architectures as a part of training (Elsken et al., 2019). However, we do not optimize the architectural properties of the network while training, instead we learn many architectures jointly.

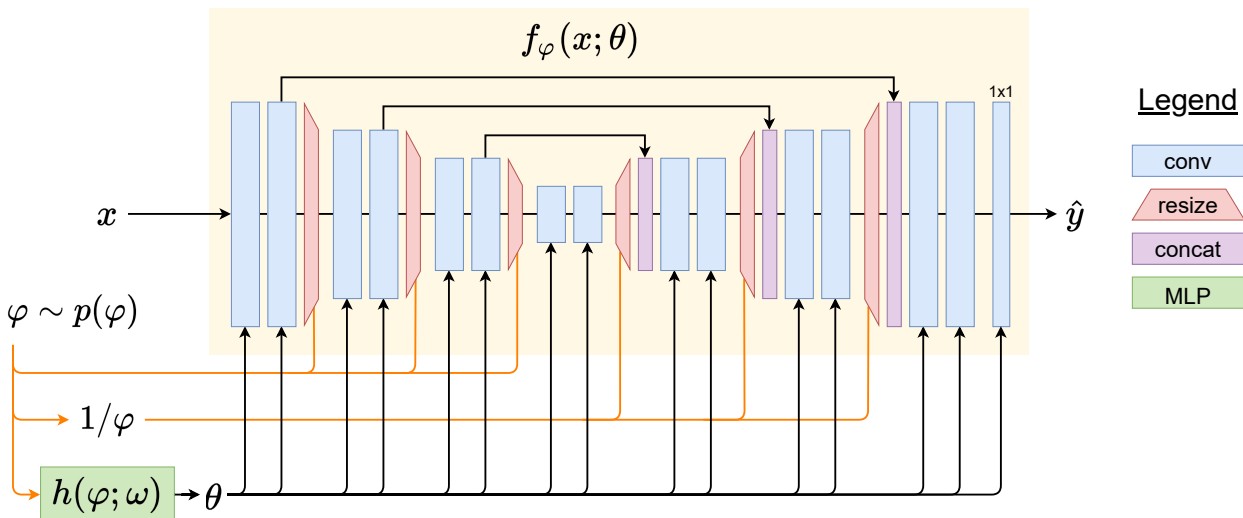

Figure 2: Diagram of the proposed hypernetwork-based model used to jointly learn a family of segmentation networks with flexible feature rescaling. At each iteration, a rescaling factor $\varphi$ is sampled from a prior distribution $p(\varphi)$. Given $\varphi$, the hypernetwork $h(\varphi; \omega)$ predicts the parameter values $\theta$ of the primary network $f_\varphi(x; \theta)$. Given a sample datapoint $x$, we predict $\hat{y} = f_\varphi(x; \theta)$ and compute the loss $\mathcal{L}(\hat{y}, y)$ with respect to the true label $y$. The resulting loss gradients only lead to updates for learnable hypernetwork weights $\omega$. The hypernetwork is implemented as a Multi Layer Perceptron (MLP) and the primary network uses a UNet-like architecture.

## 3 Method

Convolutional neural networks (CNNs) are parametric functions $\hat{y} = f(x; \theta)$ that map input values $x$ to output predictions $\hat{y}$ using a set of learnable parameters $\theta$. The network parameters $\theta$ are optimized to minimize a loss function $\mathcal{L}$ given dataset $\mathcal{D}$ using stochastic gradient descent strategies. For example, in supervised scenarios,

$$\theta^* = \arg\min_\theta \sum_{x,y \in \mathcal{D}} \mathcal{L}\left(f(x; \theta), y\right). \tag{1}$$

Most CNNs architectures use some form of rescaling layers with a *fixed* rescaling factor that rescale the intermediate network features, most often by halving (or doubling) along each spatial dimension.

We define a generalized version of convolutional models $f_\varphi(x, \theta)$ that perform feature rescaling by a *continuous* factor $\varphi \in [0, 1]$. Specifically, $f_\varphi(x; \theta)$ has the same parameters and operations as $f(x; \theta)$, with all intermediate feature rescaling operations being determined by the rescaling factor $\varphi$. Values of $\varphi = 1/2$ represent scaling by half as is prevalent in CNNs, lower values represent more aggressive rescaling, while values closer to 1 represent smaller spatial changes to the intermediate features.

Characterizing the model accuracy as a function of the rescaling factor $\varphi$ for a particular dataset using standard techniques requires training multiple instances of $f_\varphi(\cdot; \theta)$, one for each rescaling factor, leading to substantial computational burden. We propose a framework where the family of parametric functions $f_\varphi(\cdot; \theta)$ is learned jointly, using an amortized learning strategy that learns the effect of rescaling factors $\varphi$ on the network parameters $\theta$. Motivated by work on hypernetworks, we employ a function $h(\varphi; \omega)$ with learnable parameters $\omega$ that maps the rescaling ratio $\varphi$ to a set of convolutional weights $\theta$ for the function $f_\varphi(\cdot; \theta)$. We model $h(\varphi; \omega)$ with another neural network, or *hypernetwork*, and optimize its parameters $\omega$ based on an amortized learning objective

$$\omega^* = \arg\min_\omega \sum_{x,y \in \mathcal{D}} \mathcal{L}\left(f_\varphi(x; h(\varphi; \omega)), y\right), \tag{2}$$

At each training iteration, we sample a resizing ratio $\varphi \sim p(\varphi)$ from a prior distribution $p(\varphi)$, which determines both the rescaling of internal representations and, via the hypernetwork, the weights of the primary network $f_\varphi(\cdot; \theta)$. Figure 2 presents the information flow between $\varphi$ and the neural network models, where $f_\varphi(\cdot; \theta)$ illustrates the UNet-like architecture used in our experiments.

We highlight that even though the hypernetwork model has more trainable parameters than a regular primary network, the number of convolutional parameters $\theta$ is the same.

## 3.1  Rescaling Factor Selection

Once trained, the hypernetwork yields the convolutional parameters $\theta$ for any given input rescaling factor $\varphi$ in a single forward pass. We examine two different ways to take advantage of this. First, we can exploit this landscape of models to rapidly select a rescaling factor $\varphi$ under a new performance metric of interest on a held out set of data $\mathcal{D}'$ that may be different from the training dataset. For example, we focus on inference computational cost, and devise a metric $\mathcal{C}$ that captures the inference computational costs measurements, such as the number of floating point operations (FLOPs) or the peak memory consumption. We then solve the optimization problem

$$
\varphi^* = \arg\min_\varphi \; \mathcal{C}(f_\varphi(\cdot, h(\varphi; \omega), h)
$$
$$
\text{s.t.} \quad \text{Acc}(f_\varphi(\cdot, h(\varphi; \omega), \mathcal{D}') \geq \alpha,
$$
(3)

where $\text{Acc}(f_\varphi, \mathcal{D}')$ is an accuracy metric and $\alpha$ is a minimum desired accuracy. We can search for $\varphi^*$ using black box hyperparameter search strategies such as random search, grid search, or tree Parzen estimators (Bergstra & Bengio, 2012; Bergstra et al., 2011). The second way we exploit this landscape of models is to evaluate many values of $\varphi$ to fully characterize the accuracy-efficiency Pareto frontier. This can be achieved efficiently since it does not involve any further training. In both cases, once $\varphi^*$ is found, we leave out the hypernetwork entirely for inference purposes, using the predicted weights by the hypernetwork, thus incurring only the computational requirements of the primary CNN.

# 4  Experiments

We present two sets of experiments. The first set evaluates the accuracy and efficiency of segmentation models generated by our hypernetwork, both in absolute terms and relative to models generated with fixed resizing. Through the second set of experiments, we provide insights into the performance results.

## 4.1  Experimental Setup

**Datasets**. We use three widely-used segmentation datasets: OASIS, Caltech-UCSB Birds, and Oxford-IIIT Pets. OASIS is a medical imaging dataset containing 414 brain MRI scans that have been skull-stripped, bias-corrected, and resampled into an affinely aligned, common template space (Hoopes et al., 2022; Marcus et al., 2007). For each scan, segmentation labels for 24 brain substructures in a 2D coronal slice are available. We split the data into 331 images for training and validation, and 83 images for test. The Oxford-IIIT Pet dataset (Parkhi et al., 2012) comprises 7,349 natural images (3,669 for training and validation, and 3,680 for test) from 37 species of domestic cats and dogs. Caltech-UCSD Birds-200-2011 (Wah et al., 2011) consists of 11,788 images (5994 for training and validation, and 5794 for test) from 200 bird species. Both CUBBirds and OxfordPets include segmentation maps. In all datasets, we trained using an 80%-20% train-validation split of the training data.

**Segmentation Network**. We use the UNet architecture, as it is the most prevalent and widely-used segmentation CNN architecture (Ronneberger et al., 2015; Isensee et al., 2021). We use bilinear interpolation layers to downsample or upsample a variable amount based on the hyperparameter $\varphi$ for downsampling, or its inverse $1/\varphi$ for upsampling. We round the rescaled output size to the nearest integer to accommodate for discrete pixel sizes. We use five encoder stages and four decoder stages, with two convolutional operations per stage and LeakyReLU activations (Maas et al., 2013). For networks trained on OASIS Brains we use 32 features at each convolutional filter weight. For the OxfordPets and CUBBirds datasets we use an encoder

with (16, 32, 64, 128, 256) channels, and a decoder using (32, 16, 8, 4) channels at each stage respectively. We found that our results worked across various settings with many choices for the numbers of filters.

**Hypernetwork**. We implement the hypernetwork using three fully connected layers. Given a vector of $[\varphi, 1 - \varphi]$ as an input, the output size is set to the number of convolutional filters and biases in the segmentation network. We use the vector representation $[\varphi, 1 - \varphi]$ to prevent biasing the fully connected layers towards the magnitude of $\varphi$. The hidden layers have 10 and 100 neurons respectively and LeakyReLU activations (Maas et al., 2013) on all but the last layer, which has linear outputs. Hypernetwork weights are initialized using Kaiming initialization (He et al., 2015a) with *fan out*, and bias vectors are initialized to zero.

**Baseline Method**. We train a set of conventional UNet models with fixed rescaling factors. These networks are identical to the primary network, but are trained without a hypernetwork and instead use a fixed ratio of $\varphi$. We train baseline models at 0.05 rescaling factor increments for the OASIS datasets from 0 to 1, and at 0.1 increments for the OxfordPets and CUBBirds datasets from 0 to 0.6.

**Evaluation**. We evaluate each segmentation method using the Dice score (Dice, 1945), which quantifies the overlap between two regions and is widely used in the segmentation literature. Dice is expressed as

$$\text{Dice}(y, \hat{y}) = \frac{2|y \cap \hat{y}|}{|y| + |\hat{y}|},$$

where $y$ is the ground truth segmentation map and $\hat{y}$ is the predicted segmentation map. A Dice score of 1 indicates perfectly overlapping regions, while 0 indicates no overlap. For datasets with more than one label, we average the Dice score across all foreground labels.

For each experimental setting we train five replicas with different random seeds and report mean and standard deviation. Once trained, we use the hypernetwork to evaluate a range of $\varphi$, using 0.01 intervals. This is more fine-grained than is feasible for the UNet baseline because all evaluations use the same model, whereas the UNet baseline requires training separate models for each rescaling factor.

**Training**. We train the networks using a categorical cross-entropy loss. We found alternative choices of loss function, such as a soft Dice-score loss, led to similar results in all tasks. We use the Adam optimizer (Kingma & Ba, 2014) and train networks until the loss in the validation set stops improving. For networks trained on OASIS brains, we sample the rescaling hyperparameter $\varphi$ from $\mathcal{U}(0, 1)$, and for the CUBBirds and OxfordPets datasets, we sample $\varphi$ from the prior distribution $\mathcal{U}(0, 0.6)$. Because of the large number of features in the CUBBirds and OxfordPets models, we limit the prior so that the segmentation models do not exceed available GPU memory. We provide additional platform and reproducibility details in the appendix, and we will release the code for our method along with the manuscript.

### 4.2 Accuracy and Computational Cost

#### 4.2.1 Accuracy

First, we compare the accuracy of a single hypernetwork model (Hyper-UNet) to a set of baseline UNet models trained with fixed rescaling ratios. Our goal is to assess the ability of the hypernetwork model to learn a continuum of CNN weights for various rescaling factors, and evaluate how the performance of models using those weights compare to that of models trained independently with specific rescaling factors.

Figure 3 shows the Dice score on the held-out test split of each dataset as a function of the rescaling factor $\varphi$, for our approach and the baselines. We find that many rescaling factors can achieve similar final model performance, for example in the OASIS task, both strategies achieve a mean Dice score across brain structures above 0.89 for most rescaling factors. For the natural image datasets, we find that segmentation quality decreases more rapidly as $\varphi$ decreases than for OASIS.

In all datasets, the networks predicted by the hypernetwork are consistently more accurate than equivalent networks trained with specific rescaling factors. The best rescaling factor found with the hypernetwork outperformed the best ratio for the UNet baseline by 0.8 Dice points on OASIS Brains, 3.1 Dice points in the OxfordPets dataset and 5 Dice points in the CUBBirds dataset. Experiments later in the paper delve deeper into this improvement in model generalization.

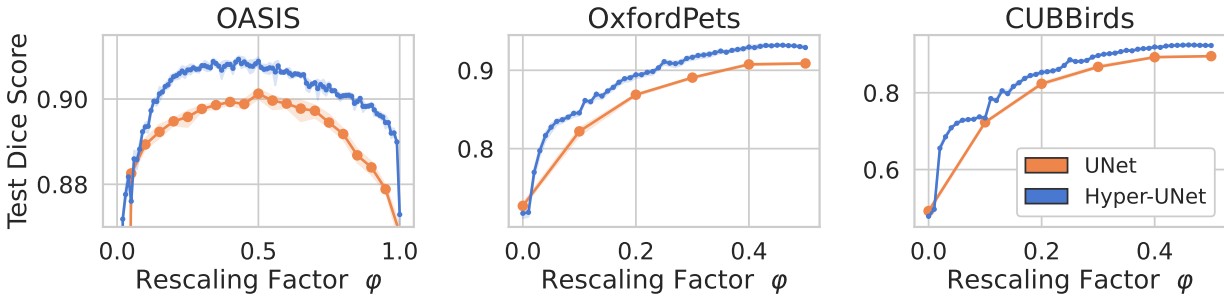

Figure 3: Comparison between a family of networks trained with fixed amounts of feature rescaling (UNet) and a single instance of the proposed hypernetwork method (Hyper-UNet) trained on the entire range of rescaling factors. We report results in the test set for the three considered segmentation datasets. In all settings the Hyper-UNet model outperforms the individually trained UNet models.

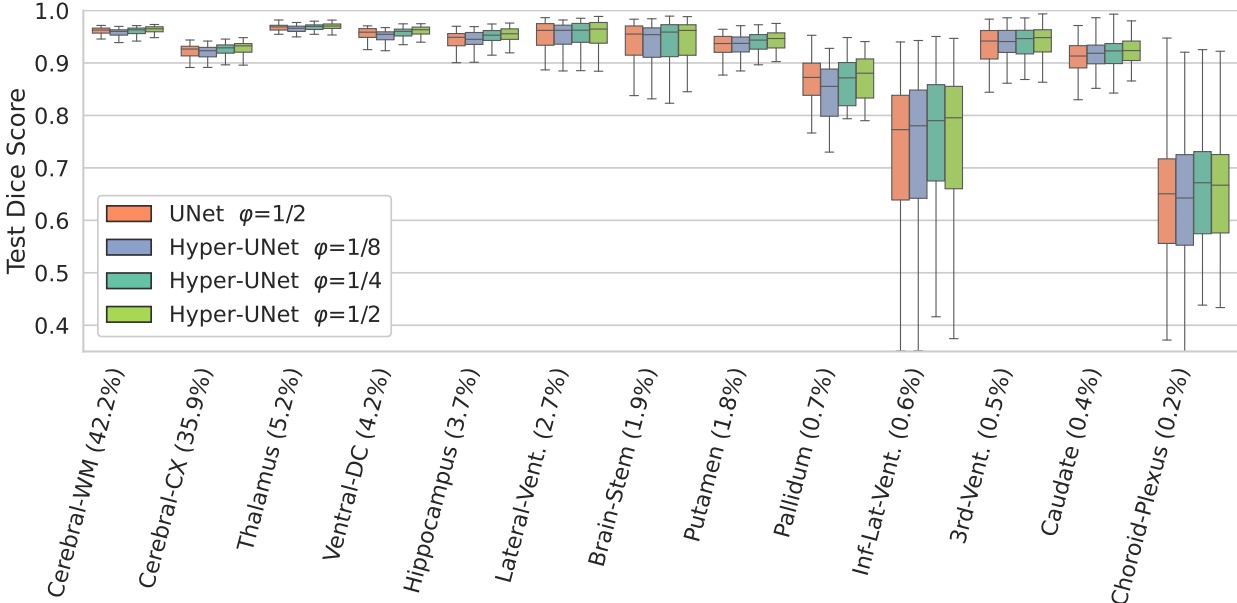

Figure 4: Segmentation Dice coefficient across various brain structures for the baseline method (UNet) trained and evaluated at $\varphi = 0.5$, and the hypernetwork approach evaluated at various rescaling factors $\varphi$ (Hyper-UNet) for the OASIS test set. Anatomical structures are sorted by mean volume in the dataset (in parentheses). Labels consisting of left and right structures (e.g. Hippocampus) are combined. We abbreviate the labels: white matter (WM), cortex (CX) and ventricle (Vent). The proposed hypernetwork model achieves similar performance for all labels despite the reduction in feature map spatial dimensions for $\varphi = 0.5$.

We also explore how varying the rescaling factor $\varphi$ affects the segmentation of smaller labels in the image. Figure 4 presents a breakdown by neuroanatomical structures for a UNet baseline with $\varphi = 0.5$ and a subset of rescaling factors for the Hypernetwork model. We observe that the proposed hypernetwork model achieves similar performance for all labels regardless of their relative size, even when downsampling by substantially larger amounts than the baseline.

### 4.2.2 Computational Cost

We analyze how each rescaling factor affects the inference computational requirements of the segmentation networks by measuring the number of floating point operations (FLOPs) required to use the network for each rescaling factor at inference time. We also evaluate the amount of time used to train each model.

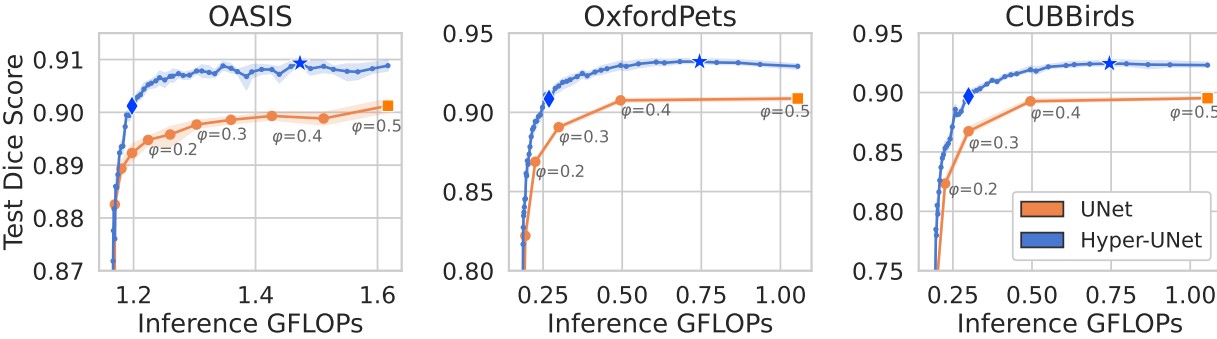

Figure 5: Trade-off curves between test model accuracy measured in Dice score and inference computational cost measured in GFLOPs for a family of networks trained with fixed amounts of feature rescaling (UNet) and for single instance of the proposed hypernetwork method (Hyper-UNet) trained on the entire range of rescaling factors. For all datasets, we find that most accurate model predicted by the hypernetwork ($\bigstar$) outperforms the best baseline network while requiring fewer inference GFLOPs. Moreover, compared to the best UNet ($\blacksquare$), our method can achieve similar results ($\blacklozenge$), at a small fraction of the computational cost. Results are averaged across network initializations and variance is indicated with shaded regions.

Figure 5 depicts the trade-off between test Dice score and the inference computational cost for both methods in all segmentation tasks. We observe that the choice of rescaling factor has a substantial effect on the computational requirements. For instance, for the OxfordPets and CUBBirds tasks, reducing the rescaling factor from the default $\varphi = 0.5$ to $\varphi = 0.4$ reduces computational cost by over 50% and by 70% for $\varphi = 0.3$. In all cases, we again find that hypernetwork achieves a better accuracy-FLOPs trade-off curve than the set of baselines. Tables 1 and 2 in the supplement report detailed measurements for accuracy (in Dice score), inference cost (in inference GFLOPs) and training time (in hours) for both approaches.

### 4.2.3 Efficiency Analysis

The hypernetwork-based approach facilitates examining the trade-off between computational efficiency and performance, as shown in Figure 5. As $\varphi$ approaches 0, accuracy sharply drops as the amount of information that the UNet can propagate becomes extremely limited. As $\varphi$ approaches 1, accuracy rapidly decreases as well, likely because with insufficient downsampling the network lacks spatial context to correctly perform the segmentation.

Because of the identical cost of inference (for a given rescaling factor) and the improved segmentation quality of the hypernetwork models, the hypernetwork method Pareto curve dominates the regular networks, requiring less inference computation for comparable model quality. For example, for networks trained on OASIS, we observe that for $\varphi \in [0.25, 0.5]$ there is no loss in segmentation quality, showing that a more than 20% reduction in FLOPS can be achieved with no loss of performance. The hypernetwork achieves similar performance to the best baseline with $\varphi = 0.15$, while reducing the computational cost of inference by 26%.

The result is even more pronounced with the Oxford-Pets dataset. At $\varphi = 0.25$ the hypernetwork model matches the accuracy of the best baseline UNet while reducing the computational cost by 76%. Furthermore, going from a rescaling factor of $\varphi = 0.5$ to $\varphi = 0.25$ offers a 4x reduction in FLOPs with a 3 Dice point decrease. Characterizing the accuracy-efficiency Pareto frontier using our method requires substantially fewer GPU-hours than the traditional approach, while enabling a substantially more fine-scale curve. In our experiments, the set of baselines required over an order of magnitude ( $10\times$) longer to train compared to the single hypernetwork model, even when we employ a coarse-grained baseline search. Had we done a more fine-grained search for the best ratio, we might have found a ratio that provided slightly better accuracy, but training would have required even more time. In practice, practitioners are rarely willing to train more than a few models. In contrast, we can instantiate our system at arbitrary values of $\varphi$ without the need to retrain.

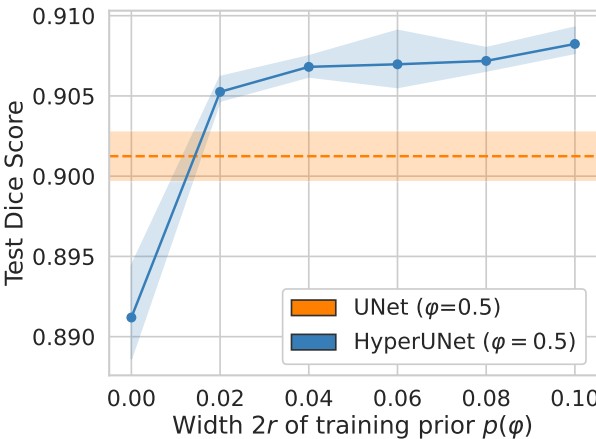
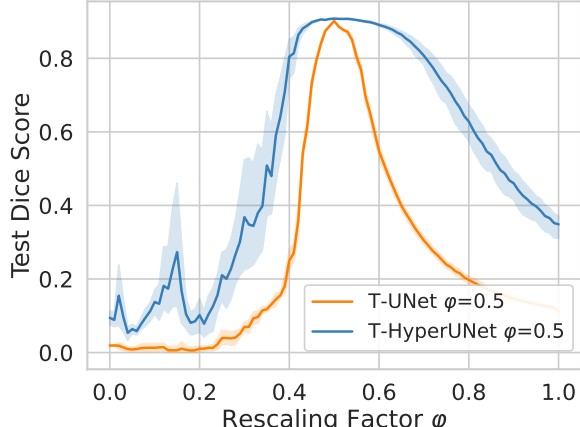

Figure 6: Test segmentation Dice on the OASIS Brains task as a function of the prior distribution range $2r$. Results are evaluated with $\varphi = 0.5$ in all settings, regardless of the training prior, and we also include a baseline UNet model. Uniform priors for $\varphi$ are of the form $\mathcal{U}(0.5 - r, 0.5 + r)$, i.e. they linearly increase around 0.5. Standard deviation (shaded regions) is computed over 5 random initializations.

Figure 7: Segmentation results for models resulting from transferring weights learned for a given rescaling factor and evaluated at different factors. The convolutional parameters learned by the hypernetwork (T-Hyper) transfer to nearby factors substantially better than the weights of the baseline model (T-UNet). Standard deviation (shaded regions) is computed over 5 random seeds.

### 4.3 Analysis Experiments

In this section we perform additional experiments designed to shed light on why the hypernetwork models achieve higher test accuracy than the baseline UNet models.

#### 4.3.1 Varying Prior Width

We first study the quality of hypernetwork segmentation results with varying width of the prior distribution $p(\varphi)$. Our goal is to understand how learning with varying amounts of rescaling affects segmentation accuracy and generalization. We train a set of hypernetwork models on the OASIS dataset but with narrow uniform distributions $\mathcal{U}(0.5 - r, 0.5 + r)$, where $r$ controls the width of the distribution. We vary $r = [0, 0.01, \ldots, 0.05]$ and evaluate them with $\varphi = 0.5$

Figure 6 shows Dice scores for training and test set on the OASIS segmentation task, as well as a baseline UNet for comparison. With $r = 0$, the hypernetwork is effectively trained with a constant $\varphi = 0.5$, and slightly underperforms compared to the baseline UNet. However, as the range of the prior distribution grows, the test Dice score of the hypernetwork models improves, surpassing the baseline UNet performance.

Figure 6 suggests that the improvement over the baselines can be at least partially explained by the hypernetwork learning a more robust model because it is exposed to varying amounts of feature rescaling. This suggests that stochastically changing the amount of downsampling at each iteration has a regularization effect on learning that leads to improvements in the model generalization. Wider ranges of rescaling values for the prior $p(\varphi)$ have a larger regularization effect. At high values of $r$, improvements become marginal and the accuracy of a model trained with the widest interval $p(\varphi) = \mathcal{U}(0.4, 0.6)$ is similar to the results of the previous experiments with $p(\varphi) = \mathcal{U}(0, 1)$.

#### 4.3.2 Weight transferability.

We study the effect on segmentation results associated with using a set of weights that was trained with a different rescaling factor. Our goal is to understand how the hypernetwork adapts its predictions as the rescaling factor changes. We use *fixed* weights $\theta' = h(0.5; \omega)$ as predicted by the hypernetwork with

input $\varphi' = 0.5$, and run inference using different rescaling factors $\varphi$. For comparison, we apply the same procedure to a set of weights of a UNet baseline trained with a fixed $\varphi = 0.5$.

Figure 7 presents segmentation results on OASIS for varying rescaling factors $\varphi$. Transferring weights from the baseline model leads to a rapid decrease in performance away from $\varphi = 0.5$. In contrast, the weights generated by the hypernetwork are more robust and can effectively be used in the range $\varphi \in [0.45, 0.6]$. Nevertheless, weights learned for a specific rescaling factor $\varphi$ do not generalize well to a wide range of different rescaling factors, whereas weights predicted by the hypernetwork transfer substantially better to other rescaling factors.

## 5  Limitations

The proposed framework is designed for and evaluated on image segmentation, a prevalent task in computer vision and medical image analysis. We believe that it can be extended to other tasks like image registration or object detection. Constructing such extensions is an exciting area for future research.

When including rescaling factors $\varphi > 0.5$ as part of the prior, the proposed method has higher peak memory requirements than a regular UNet baseline (with $\varphi = 0.5$), which might require GPUs with larger memory capacity or model-parallel distributed training. The hypernetwork also introduces additional hyperparameters related to its architecture, such as the number of hidden neurons in each layer and the number of layers. While this is a complication, in our experiments, we found that many settings for these additional parameters led to similar results as those presented in the results.

Even though the proposed method is applicable to any convolutional neural network, we demonstrate it only on UNet architectures. We chose the UNet as a benchmark architecture because it is a popular choice for segmentation networks across many domains (Ronneberger et al., 2015; Drozdzal et al., 2016; Isola et al., 2017; Isensee et al., 2018; Balakrishnan et al., 2019; Billot et al., 2020). It also features key properties, such as an encoder-decoder structure and skip connections from the encoder to the decoder, that are used in other segmentation architectures (Chen et al., 2017; Zhao et al., 2017; Chen et al., 2018).

## 6  Summary

We introduce a hypernetwork-based approach for efficiently learning a family of CNNs with different rescaling factors in an amortized way.

Given a trained model, this proposed approach enables powerful downstream applications, such as efficiently generating the Pareto frontier for the trade-off between inference cost and accuracy. This makes it possible to rapidly search for a rescaling factor that maximizes model a new metric of interest at inference without retraining, such as maximizing accuracy subject to computational cost constraints. For example, using the hypernetwork, we demonstrate that using larger amounts of downsampling can often lead to a substantial reduction in the computational costs of inference, while sacrificing nearly no accuracy. Finally, we demonstrate that for a variety of image segmentation tasks, models learned with this method achieve accuracy at least as good, and more often better than, that of CNNs trained with fixed rescaling ratios, while producing more robust models. This strategy enables the construction of a new class of cost-adjustable models, which can be adapted to the resources of the user and minimize carbon footprint while providing competitive performance. We believe such models will substantially enable deployment of current machine learning systems across a variety of domains.

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
