# OpenReview forum: "Amortized Learning of Flexible Feature Scaling for Image Segmentation"
_TMLR — Rejected by TMLR_

### Review · Reviewer_AF5F · 2023-03-10

**Summary Of Contributions:**

This paper addresses the problem of using neural networks at multiple feature dimensions. Such trade-offs are necessary to switch between an environment where compute is fully available (data centre), and an environment where compute is restricted (embedded, on-device). This paper studies this problem from the perspective of hypernetworks, where a neural network outputs the parameters of another neural network. In this case, the paper amortises the feature scaling parameter. This parameter can interpolate between no spatial reduction on one side, and the factors of 2 downsizing that existing approaches use.
The proposed method saves computing compared to a full hyperparameter sweep, where multiple models need to be trained.

However, the experimental results lack comparisons, which make them hard to compare. Lots of emphasis is given to the OASIS dataset, Figure 3, 4, 6, and 7. However, no established results on this dataset are compared against. Figure 5 shows an improvement from .9 to .91 dice score, but it’s hard to say if any of these numbers are state-of-the-art.


**Audience:**

Yes

**Broader Impact Concerns:**

n.a. (proposed method will enable evaluation on more evaluation settings.)

**Claims And Evidence:**

No

**Requested Changes:**

Do I understand correctly that this paper has no appendix? I would have expected a few paragraphs explaining details about the following:

1) The paper needs more clarification on the resizing method: how are continuous values for rescaling achieved for a discrete-domain grid (images)? Also, do the amount of parameters scale accordingly; in other words, does the output of h(\phi, \omega) vary as a function of \phi? The authors briefly cover this topic with the following sentence, but more explanation is needed to ensure reproducible results. (Quote from the paper ‘... round the rescaled output size to the nearest integer to accommodate for discrete pixel sizes.’)
2) How is the actual rescaling performed on the fixed-size feature maps? Would the output of h() determine the rescaling, or is rescaling done with a differential but non-parameterised resizing layer?

Smaller questions:
* How does the proposed method compare with YOTO [1]?
* How does such an approach compare with kernels in the continuous domain, e.g. [2]?
* Figure 4, ‘brain structures’ are subsets of the OASIS dataset?
* Figure 6, how is the errorbar for UNet obtained? If I understand correctly the original UNet does not have a dynamic rescaling. Does the indicated noise result from the use of random initialisations?
* Figure 7 shows distinct peaks. Does this have to do with artefacts in the rescaling, e.g. aliasing?


[1] Dosovitskiy and Djolonga. "You only train once: Loss-conditional training of deep networks." International conference on learning representations. 2020.

[2] Finzi, Bondesan, and Welling. "Probabilistic numeric convolutional neural networks." arXiv preprint arXiv:2010.10876 (2020).

[3] Huang, Lang, et al. "Interlaced sparse self-attention for semantic segmentation." arXiv preprint arXiv:1907.12273 (2019).

[4] Yun, Sangdoo, et al., "Cutmix: Regularization strategy to train strong classifiers with localizable features." ICCV  2019.


**Strengths And Weaknesses:**

Strengths:
  * The proposed method allows continuous interpolation of scaling factors. However, it’s hard to say if this idea is novel, see [1].
  *  The paper makes a comparison in terms of scaling and compute (Figure 6).



Weaknesses:

  * The main weakness of this paper is its experimental validation. None of the three datasets are compared to established results. For both OASIS, Oxford Pets and CUB200 the paper does not mention results obtained by other papers.
For example, in Figure 4 the results of UNet and Hyper-UNet overlap a lot. It’s hard to say if a) these results are state-of-the art, and b) if the improvement is significant.


  * Figure 3 shows a worse result on OASIS dice score for fewer rescaling. This observation seems strange and requires an explanation. In contrast, figures (b) and (c) do show the expected relation between increasing dice score for increasing resolution (decreasing downscaling). How can this phenomenon be explained, and if this is particular for the currently used regularisation strategy, then how much?

* The caption of Figure 5 mentions test accuracy. However, I understood from Section 4.1 that this paper considers segmentation tasks. I assume this is a typo.

---

> ### Author Response · Authors · 2023-03-27
> **Reply to Reviewer AF5F**
>
> We are thankful for the constructive feedback and comments. Addressing the raised questions:
>
> > The main weakness of this paper is its experimental validation. None of the three datasets are compared to established results. For both OASIS, Oxford Pets and CUB200 the paper does not mention results obtained by other papers. For example, in Figure 4 the results of UNet and Hyper-UNet overlap a lot. It’s hard to say if a) these results are state-of-the art, and b) if the improvement is significant.
>
> -   **Comparison to other works**. Our goal with this work was not to
>     beat the state-of-the-art segmentation results for the evaluated
>     datasets, but to demonstrate the viability of using an amortized
>     learning strategy to rapidly explore the range of possible rescaling
>     factors in segmentation networks. Our results show that this
>     strategy is possible and effective, and that it can replace a
>     hyperparameter search involving training many models. Specifically,
>     we show that the models produced by our hypernetwork are never worse
>     than the corresponding family of baseline models in terms of
>     segmentation results, proving the utility of our method. Moreover,
>     before evaluating our method, we performed a thorough hyperparameter
>     search for the baseline models on each task to find the best setting. We evaluate our method
>     using that same hyperparameter setting (model architecture, data
>     augmentation, optimizer, etc) without further tuning.
>
> > Figure 3 shows a worse result on OASIS dice score for fewer rescaling. This observation seems strange and requires an explanation. In contrast, figures (b) and (c) do show the expected relation between increasing dice score for increasing resolution (decreasing downscaling). How can this phenomenon be explained, and if this is particular for the currently used regularisation strategy, then how much?
>
> -   **Worse results with higher rescaling factor $\varphi$**. While
>     increasing the rescaling factor $\varphi$ increases the resolution,
>     it simultaneously decreases the amount of global context that the
>     network can effectively aggregate because the number of layers in
>     the network stays fixed. For instance, with $\varphi = 1.0$ the
>     spatial dimensions stay constant throughout the network, so all
>     convolutions are performed at the input resolution, which leads to
>     worse results. In practice, most convolutional architectures
>     downscale the image repeatedly to aggregate context at multiple
>     scales. Lastly, figures (b) and (c) do not show this behaviour since
> for those experiments we had to constrain $\varphi \in [0,0.5]$ due to GPU memory constraints.
>
>
> > Do I understand correctly that this paper has no appendix?
>
> -   **Missing Appendix**. We included supplementary material as part of
>     the submission, and it is available at the following OpenReview link
>     (<https://openreview.net/attachment?id=nPwdnRNJEP&name=supplementary_material>).
>
> > The paper needs more clarification on the resizing method: how are continuous values for rescaling achieved for a discrete-domain grid (images)? Also, do the amount of parameters scale accordingly; in other words, does the output of h(\phi, \omega) vary as a function of \phi? The authors briefly cover this topic with the following sentence, but more explanation is needed to ensure reproducible results. (Quote from the paper ‘... round the rescaled output size to the nearest integer to accommodate for discrete pixel sizes.’)
>
> -   **Continuous Size on Discrete Images**. Inputs to the
>     hypernetwork are continuous. Resizing layers take the continuous
>     value of $\varphi$, multiply it by the input image size, round
>     to the nearest integer for each dimension, and use bilinear
>     interpolation to rescale to this target size.
>
> -   **Number of parameters**. The number of parameters $\theta$
>     predicted by $h(\varphi;\omega)$ is the same for all $\varphi$
>     values as the number of convolutional parameters does not
>     change. We implement the hypernetwork with an MLP architecture,
>     and for a U-Net with $N$ parameters, our MLP has three fully
>     connected layers of sizes (1 $\times$ 10), (10 $\times$ 100) and
>     (100 $\times$ N). So $h(\varphi;\omega)$ takes one input, the
>     rescaling factor, and predicts all convolutional
>     parameters $\theta$.
>
> > How is the actual rescaling performed on the fixed-size feature maps? Would the output of h() determine the rescaling, or is rescaling done with a differential but non-parameterised resizing layer?
>
> -   **Rescaling implementation**. The rescaling is done using a
>     differentiable but non-parametrised resizing layer. In particular,
>     we use a bilinear interpolation strategy to go from $H \times W$
>     spatial dimensions to
>     $\text{round}(\varphi H) \times \text{round}(\varphi W)$.

---

> ### Author Response · Authors · 2023-03-27
> **Reply to Reviewer AF5F (continued)**
>
> Smaller questions:
>
> > How does the proposed method compare with YOTO [1]?
>
> 1.  The YOTO\[1\] authors explore solving many problems jointly by
>     sampling loss parameters and conditioning the model on these. In
>     contrast, we study learning multiple architectures with different
>     rescaling factors jointly. Additionally, our proposed model design
>     also differs. The YOTO authors use FiLM (Feature-wise Linear
>     Modulation) layers to condition the model, which apply pixelwise
>     affine transformations to modulate intermediate feature maps. We
>     cannot use this approach, as different rescaling factors will lead
>     to feature maps of different spatial dimensions. In contrast, we use
>     a hypernetwork that predicts the convolutional parameters directly.
>     Related work suggests that hypernetworks are more modular and
>     parameter efficient than conditioning approaches \[5\]. We will
>     include this discussion in the revision.
>
> > How does such an approach compare with kernels in the continuous domain, e.g. [2]?
>
> 2.  Our proposed method differs from \[2\] because we do not attempt to
>     learn models that operate in the continuous domain, but rather learn
>     a hypernetwork that models a continuum of models, each of which
>     operate in the discrete pixel domain. When performing the rescaling
>     operations, the scale factor is not differentiated as we do not
>     optimize it, instead we sample it from a prior. At inference time,
>     our approach allows selecting a discrete model for a given scale
>     factor $\varphi$.
>
> > Figure 4, ‘brain structures’ are subsets of the OASIS dataset?
>
> 3.  In Figure 4, the brain structures correspond to segmentation labels
>     available in the OASIS dataset.
>
> > Figure 6, how is the errorbar for UNet obtained? If I understand correctly the original UNet does not have a dynamic rescaling. Does the indicated noise result from the use of random initialisations?
>
> 4.  In Figure 5, the error bar is computed across 5 models trained with
>     different random initializations.
>
> > Figure 7 shows distinct peaks. Does this have to do with artefacts in the rescaling, e.g. aliasing?
>
> 5.  In Figure 7, we believe that the peaks do not correspond to specific
>     artifacts in the rescaling as they only occur for the hypernetwork
>     model and not for the regular model. Moreover, not all repetitions
>     of the hypernetwork present the same peaks, with different random
>     initializations presenting peaks at different $\varphi$ values.
>
> \[5\] On the Modularity of Hypernetworks, Galanti et al. NeurIPS 2020

---

### Review · Reviewer_mqq7 · 2023-03-11

**Summary Of Contributions:**

The paper proposes a segmentation model whose inference cost (and resulting accuracy) can be adjusted at inference time. The model is a UNet for semantic segmentation. The dynamic inference cost works by using a hypernetwork during training. The hypernetwork predicts the parameters of the downsampling layers in the UNet as a function of a desired downsampling ratio. At inference time, the downsampling ratio can be adjusted to trace the cost/accuracy frontier.

**Audience:**

No

**Broader Impact Concerns:**

No concerns.

**Claims And Evidence:**

No

**Requested Changes:**

The most important changes requested are the following:
* Limit the claims to UNet and semantic segmentation (or include other networks and/or segmentation tasks).
* Include Pascal VOC and Context, and ideally ADE20k, and include reference performance numbers from the literature.
* Include the variable resolution baseline.

**Strengths And Weaknesses:**

Strengths

* The method provides a clean/elegant solution for dynamically adjusting the cost/accuracy trade-off at inference time.
* The results indicate that the model can be used to successfully increase or decrease FLOPS, with according changes in performance on semantic segmentation.
* The results indicate the the model performs better, even at any given inference cost (i.e. fixed downsampling factor), than a regular network trained with at that fixed downsampling factor.
* It is faster to train a single hypernetwork than multiple networks with different downsampling factors. GPU-hours are reported in the Supplementary, but I think could be included in the main text also.

Weaknesses

* The abstract/introduction indicates that the method works for CNNs for segmentation. However, evaluation is only performed on semantic segmentation (as opposed to instance/panoptic), and the model is only instantiated using a UNet. The abstract and introduction therefore seem to over-claim the generality of the method supported by the experimental evidence.
* Non-standard evaluation tasks are used, and no reference performance scores from the literature are provided. Two of the three datasets (Pets and CUB) are more usually used for evaluation of fine-grained image-classification. To contextualize the results, and assess the validity of the baseline, it is essential to evaluate on some datasets more widely used in semantic segmentation: e.g. Pascal VOC, Pascal Context, and ADE20k. If computational resources are a bottleneck, Pascal is a small dataset, of similar size to, or smaller than, those already in the paper.
* There is an important baseline missing, that would greatly increase potential interest in the paper if it were included. One could train the UNet on multiple image resolutions and adjust the image resolution at inference time. This would offer a similar FLOPS/performance trade-off, and it would be good to know if the hypernetwork approach improves on this simple baseline.
* The details of the hypernetwork are missing. Could the authors clarify precisely the layers generated by the network.
* (minor) It is impressive that the model outperforms the baseline at fixed downsampling factor, and evidence is provided for a regularization effect. However, it would be useful to understand how this would interact with stronger data augmentation. The supplementary states that cropping/flipping is used, however, it would be useful to know the details of these augmentations and evaluate with stronger augmentations.
* (minor) The introduction of downsampling is a little imprecise "In practice, most network designs downscale by a factor of two". Could the authors disentangle the notions of downsampling stages, the overall downsampling factor of the network, and how downsampling layers are implemented in typical networks.

---

> ### Author Response · Authors · 2023-03-28
> **Reply to  Reviewer mqq7**
>
> We are thankful for the constructive feedback and comments. Addressing
> the raised questions:
>
> > The abstract/introduction indicates that the method works for CNNs for segmentation. However, evaluation is only performed on semantic segmentation (as opposed to instance/panoptic), and the model is only instantiated using a UNet. The abstract and introduction therefore seem to over-claim the generality of the method supported by the experimental evidence.
>
> -   **Limitation to UNet networks**. We designed our solution for
>     segmentation tasks, where the U-Net is the predominant architecture
>     and the large majority of alternative networks are derived from the
>     U-Net architecture. Motivated by the reviews, we tested our method
>     using other successful segmentation architectures, including FPN and
>     PSPNet. The results are consistent with the findings in
>     the U-Net architecture -- the hypernetwork model learns weights for
>     the whole range of rescaling factors at a fraction of the training
>     time, without sacrificing accuracy. Figures and full experimental
>     details are available at this anonymized
>     [link](https://raw.githubusercontent.com/anonresearcher9/tmlr871/main/arch.pdf).
>     We will include these results in the revised version.
>
>
> > Non-standard evaluation tasks are used, and no reference performance scores from the literature are provided. Two of the three datasets (Pets and CUB) are more usually used for evaluation of fine-grained image-classification. To contextualize the results, and assess the validity of the baseline, it is essential to evaluate on some datasets more widely used in semantic segmentation: e.g. Pascal VOC, Pascal Context, and ADE20k. If computational resources are a bottleneck, Pascal is a small dataset, of similar size to, or smaller than, those already in the paper.
>
> -   **Additional Semantic Segmentation Benchmarks**. We chose OxfordPets
>     and CUBBirds tasks as they present a large variance of perspectives,
>     backgrounds and segmentations masks, while not being as
>     computationally demanding, because we needed to train many models for
>     the fixed scale baselines. We agree that evaluating on additional semantic
>     segmentation datasets would make the results more robust. We are
>     actively working on training models on the proposed datasets and we
>     will post an update as soon as we have results.
>
> > There is an important baseline missing, that would greatly increase potential interest in the paper if it were included. One could train the UNet on multiple image resolutions and adjust the image resolution at inference time. This would offer a similar FLOPS/performance trade-off, and it would be good to know if the hypernetwork approach improves on this simple baseline.
>
> -   **Variable Resolution Baseline**. We thank the reviewer for
>     suggesting this additional baseline trained on variable resolutions.
>     We performed experiments in the OASIS dataset and found that this
>     baseline cannot match the performance of the proposed hypernetwork nor the individually trained
>     UNets. We include full experimental details and results in this
>     anonymized
>     [link](https://raw.githubusercontent.com/anonresearcher9/tmlr871/main/multires.pdf).
>     We will include these results in the final version.
>
> > The details of the hypernetwork are missing. Could the authors clarify precisely the layers generated by the network.
>
> -   **Hypernetwork details**. In our experiments the hypernetwork
>     predicts all the parameters of the primary network. In the revised
>     version, we expand the hypernetwork details to include this
>     specification, along with a more detailed architectural description.
>
> > (minor) It is impressive that the model outperforms the baseline at fixed downsampling factor, and evidence is provided for a regularization effect. However, it would be useful to understand how this would interact with stronger data augmentation. The supplementary states that cropping/flipping is used, however, it would be useful to know the details of these augmentations and evaluate with stronger augmentations.
>
> -   **Data augmentation**. We explored using stronger augmentations during our hyperparemeter search,
>     including affine transformations and color jitter, but found that
>     there was no significant improvement to the baselines when including
>     those.
>
> > (minor) The introduction of downsampling is a little imprecise "In practice, most network designs downscale by a factor of two". Could the authors disentangle the notions of downsampling stages, the overall downsampling factor of the network, and how downsampling layers are implemented in typical networks.
>
> -   **Downsampling**. We will clarify this. We meant that downscaling
>     and upscaling by a factor of two is a common operation in CNNs to aggregate spatial context.

---

### Review · Reviewer_6ky4 · 2023-03-14

**Summary Of Contributions:**

This work introduces the hypernetwork strategy to generate CNN parameters by given a rescaling factor. The attempt is interesting and meaningful. A continuous range of rescaling ratios can be covered by only training a single hypernetwork.
Several experiments  demonstrate the method enables powerful downstream applications. An interesting effect brings by hypernetwork-based weights generation is that the loss of performance is small. Moreover, this work constructs the flexible downsample and upsample scaling factors which is different form fixed step size pooling and upsampling.


**Audience:**

Yes

**Broader Impact Concerns:**

No.

**Claims And Evidence:**

No

**Requested Changes:**

1. The writing of this paper need to be amended, including the weaknesses and the contributions in the Introduction part.
2. Add more experiments on other latest models. It is better to extend this method to some other fields, such as image classification, object recognition.

**Strengths And Weaknesses:**

Strengths:
1. The idea is interesting and provides another approach for minimizing computational cost.
2. By using hypernetwork, many architectures can be learned jointly, which is an alternative to train multi-networks separately and might be an inspiration to other researchers.
3. The hypernetwork-based model seems to be more robust because it is exposed to multi-scale features, but with less computational cost compared with other multi-scale models.
4. The experiments on UNet show that the method is valid.

Weaknesses:
1. The performance and efficiency analysis are only applied on UNet, which is thin and insufficient. It will be better to evaluate the performance on latest models and extend to other fields, e.g. classification or object recognition.
2. This method is very interesting. Is the parameter generation MLP model is sufficient to produce appropriate weights? Because the Unet has too many parameters comparing with the generation model.
2. The writing style is not clear and scientific enough, for example, in 3.1 rescaling factor selection, it’s confusing to explain your specific methods by “We can search for”. And in 4.3.1, varying amounts of feature rescaling.
3. Some sentences are obvious and convey no additional information, e.g. reducing the rescaling factor from the default φ = 0.5 to φ = 0.4 reduces computational cost by over 50% and by 70% for φ = 0.3, in 4.2.2.
4. Some nouns are miswritten, such as FLOPs, miswritten by FLOPS in 4.2.3.

---

> ### Author Response · Authors · 2023-03-27
> **Reply to Reviewer 6ky4**
>
> We are thankful for the constructive feedback and comments. Addressing
> the raised questions:
>
> > The performance and efficiency analysis are only applied on UNet, which is thin and insufficient. It will be better to evaluate the performance on latest models and extend to other fields, e.g. classification or object recognition.
>
> > Add more experiments on other latest models. It is better to extend this method to some other fields, such as image classification, object recognition.
>
> -   **Limitation to UNet networks**. We designed our solution for
>     segmentation tasks, where the U-Net is the predominant architecture
>     and the large majority of alternative networks are derived from the
>     U-Net architecture. Motivated by the reviews, we tested our method
>     using other successful segmentation architectures, including FPN,
>     and PSPNet. The results are consistent with the findings in
>     the U-Net architecture -- the hypernetwork model learns weights for
>     the whole range of rescaling factors at a fraction of the training
>     time, without sacrificing accuracy. Figures and full experimental
>     details are available at this anonymized
>     [link](https://raw.githubusercontent.com/anonresearcher9/tmlr871/main/arch.pdf).
>     We will include these results in the revised version.
>
> -   **Other Applications**. We agree that extensions to other
>     applications are interesting avenues for future research. However,
>     image segmentation is an important and widely tackled problem across
>     both computer vision and medical image analysis, and we believe
>     demonstrating impact in image segmentation is a significant
>     contribution.
>
> > This method is very interesting. Is the parameter generation MLP model is sufficient to produce appropriate weights? Because the Unet has too many parameters comparing with the generation model.
>
> -   **MLP Architecture**. The MLP is sufficient for the parameter
>     generation. This is because it has substantially more parameters
>     than the U-Net. For a U-Net with $N$ parameters, our MLP has three
>     fully connected layers of sizes (1 $\times$ 10), (10 $\times$ 100)
>     and (100 $\times$ N). In practice, this means that in our
>     experiments, the hypernetwork has over a 100 times more learnable
>     parameters than the U-Net. Nevertheless, the overhead introduced by
>     the hypernetwork is heavily amortized because of the substantial
>     weight reuse of the predicted convolutional filters.
>
> > The writing style is not clear and scientific enough [...]
>
> >  The writing of this paper need to be amended [...]
>
> -   **Writing**. We will address the presentation and writing comments
>     in the revised version of the paper and we thank the reviewer for bringing them to our attention

---

### Review · Reviewer_pXm5 · 2023-03-19

**Summary Of Contributions:**

This paper discussed the adaptive down/up-scale factors in the UNet-based semantic segmentation network. Typically, CNN segmentation architectures resize spatial dimensions by a fixed factor of two, but recent work has explored other factors for better accuracy/efficiency. The authors show that a single hypernetwork can generate CNN parameters based on the rescaling factor, allowing for quick determination of the optimal trade-off between accuracy and efficiency. The hypernetwork outperforms CNNs with fixed rescaling factors and is useful for various domains.


**Audience:**

Yes

**Broader Impact Concerns:**

No concerns

**Claims And Evidence:**

Yes

**Requested Changes:**

Overall, this is a high-quality submission, and I am leaning toward accepting it. And I'd like to read the authors' responses to the above questions, which are essential as well.

**Strengths And Weaknesses:**

Pros/Cons:
+ The proposed idea is novel and interesting. The whole pipeline looks like a semi-AutoML framework with the inputs of resize factors for hypernet to generate weights for CNN. And such resize factors can be decided by certain efficiency/accuracy metrics.
+ There are solid experiments conducted to verify the effectiveness of the proposed method. It has outperformed the baselines with the fixed resize factors.
+ This paper is well-written, with clear logic and a comprehensive discussion of related literature.
+ Despite the promising results and innovative ideas, the solution seems to be limited within the Unet-based network, which cannot be easily adopted to other architectures. This has largely limited the usage and potential impact of this work.
+ The details of the hypernet are not clear. What is the exact dimension of hypernet layers, and how to align the output of hypernet to CNN weights? Is it any post-processing steps?
+ In AutoML, the training cost is another concern. What is the training cost of hypernet? And how much extra computation cost (compared with training a CNN) is needed for this?

---

> ### Author Response · Authors · 2023-03-27
> **Reply to Reviewer pXm5**
>
> We are thankful for the constructive feedback and comments. Addressing
> the raised questions:
>
> > Despite the promising results and innovative ideas, the solution seems to be limited within the Unet-based network, which cannot be easily adopted to other architectures. This has largely limited the usage and potential impact of this work.
>
> -   **Limitation to UNet networks**. We designed our solution for
>     segmentation tasks, where the U-Net is the predominant architecture
>     and the large majority of alternative networks are derived from the
>     U-Net architecture. Nevertheless, our method can be extended to
>     other architectures. Given an arbitrary segmentation network, our
>     method requires replacing fixed resizing layers with variable
>     resizing layers and using the hypernetwork to predict all the
>     convolutional parameters of the network. Motivated by the reviews,
>     we tested our method using other successful segmentation
>     architectures, including FPN and PSPNet. The results are consistent
>     with the findings in the U-Net architecture -- the hypernetwork
>     model learns weights for the whole range of rescaling factors at a
>     fraction of the training time, without sacrificing accuracy. Figures
>     and full experimental details are available at this anonymized
>     [link](https://raw.githubusercontent.com/anonresearcher9/tmlr871/main/arch.pdf).
>     We will include these results in the revised version.
>
> > The details of the hypernet are not clear. What is the exact dimension of hypernet layers, and how to align the output of hypernet to CNN weights? Is it any post-processing steps?
>
> -   **Hypernetwork dimensions**. Our hypernetwork is implemented using a
>     fully connected network with two hidden layers of 10 and 100 neurons
>     respectively. For each weight tensor in the primary network, we
>     learn a fully connected layer. For example, for a convolutional
>     layer with 16 input channels and 32 output channels, with weight
>     matrix of size ($16\times32\times3\times3$ = 4608), we have a
>     learnable fully connected layer of 100 input dimensions and 4608
>     output dimensions, and then reshape the output tensor to
>     $16\times32\times3\times3$. There are no postprocessing steps to
>     either the output of the hypernetwork or the primary network. We
>     will clarify our hypernetwork architectural details further in the
>     final version.
>
> > In AutoML, the training cost is another concern. What is the training cost of hypernet? And how much extra computation cost (compared with training a CNN) is needed for this?
>
> -   **Training time**. Training time for all networks is included in
>     Section A of our appendix. Overall, we found that training a single
>     hypernetwork-based model took around 1.5 times longer than training
>     a single U-Net model with the default setting of $\varphi=0.5$.
>     First, we found that the hypernetwork model converged at a similar
>     rate to the fixed scale baseline models despite the increased
>     complexity of the learning task. Training cost per iteration is
>     mainly determined by the rescaling factor. The larger the value
>     of $\varphi$, the less downscaling is applied to the intermediate
>     activations and the larger the number of FLOPS required to perform
>     the convolutions. For the OASIS task, the baseline models range from
>     0.26h (for a rescaling factor $\varphi$ = 0.05) to 0.86h (when
>     $\varphi$ = 0.95). Since the hypernetwork model uniformly samples
>     $\varphi$ from \[0, 1\], we expect the model to require roughly the
>     average over the range (approx 0.5h), plus an additional overhead
>     from the computation associated with the hypernetwork module, which
>     leads to the 0.64h runtime. The overhead of the hypernetwork is
>     heavily amortized because of the substantial weight reuse of
>     convolutional filters.

---

### Decision · Action_Editors · 2023-04-24

**Recommendation:** Reject

**Comment:**

The AE listed the reasoning behind the decision. Please refer to the claims and evidence above.

**Audience:**

This paper may be interested to many researchers, working for various topics, such as image segmentations, hyper-network, and NAS.

**Claims And Evidence:**

This paper proposes a new method to generate flexible feature scaling models and the parameters are generated by a small model. Such an idea of hypernet is novel and inspiring. This paper is reviewed by four experts on this topics; and three of them have negative final scores to the paper. The reviewers make very good and extensive comments and suggestions to improve this paper.

In general, the key problems lie in the over-claimed contributions, writing vague, and limited empirical evaluations. Particularly,
 (1)over-claimed contributions: it only addresses semantic segmentation. Especially, the authors show that the methods are in good favor of medical image segmentation.  Since the paper claims a general solution for fixed resolution testing in semantic segmentation, the reviewers suggested that it is reasonable and natural to do more  evaluation on multiple datasets and model architectures. Despite some additional results are added, it may be not enough to support the claims in general.
(2)writing vague e.g. "Most CNN segmentation architectures resize spatial dimensions by a fixed factor of two to aggregate spatial context."
(3) limited empirical evaluations: No standard evaluation is provided (like Pascal VOC) so the results cannot be confirmed against the literature. It shall also be further considered to add more results on  large benchmark dataset and other vision tasks, as suggested.

So the reviewers are worrying that there may be not enough evidence supports the claims. To address all of the points above, it may demand significant efforts in improving the paper.  So the AE may suggest a thorough major revision to this paper, and can not accept it at this time.